# Artificial Gravity Attenuates the Transcriptomic Response to Spaceflight in the Optic Nerve and Retina

**DOI:** 10.3390/ijms252212041

**Published:** 2024-11-09

**Authors:** Isaac Kremsky, Reyna Pergerson, Stephen Justinen, Seta Stanbouly, Jeffrey Willey, Charles A. Fuller, Satoru Takahashi, Martha Hotz Vitaterna, Mary Bouxsein, Xiaowen Mao

**Affiliations:** 1Department of Basic Sciences, Division of Biomedical Engineering Sciences (BMES), Loma Linda University School of Medicine, Loma Linda, CA 92350, USA; ikremsky@llu.edu (I.K.); rpergerson@students.llu.edu (R.P.); sjustinen@students.llu.edu (S.J.); sstanbouly@llu.edu (S.S.); 2Center for Genomics, Loma Linda University School of Medicine, Loma Linda, CA 92350, USA; 3Department of Radiation Oncology, Wake Forest University School of Medicine, Winston-Salem, NC 27101, USA; jwilley@wakehealth.edu; 4Department of Neurobiology, Physiology & Behavior, University of California, Davis, One Shields Avenue, Davis, CA 95616, USA; cafuller@ucdavis.edu; 5Laboratory Animal Resource Center in Transborder Medical Research Center, Department of Anatomy and Embryology, Institute of Medicine, University of Tsukuba, Tsukuba 305-8575, Ibaraki, Japan; satoruta@md.tsukuba.ac.jp; 6Center for Sleep and Circadian Biology, Department of Neurobiology, Northwestern University, Evanston, IL 60208, USA; m-vitaterna@northwestern.edu; 7Center for Advanced Orthopedic Studies, Beth Israel Deaconess Medical Center, Department of Orthopedic Surgery, Harvard Medical School, Boston, MA 02215, USA; mbouxsei@bidmc.harvard.edu

**Keywords:** spaceflight associated neuro-ocular syndrome (SANS), optic nerve, retina, RNA-seq, artificial gravity

## Abstract

The development of eye pathology is a serious concern for astronauts who spend time in deep space. Microgravity is a major component of the spaceflight environment which could have adverse effects on ocular health. The use of centrifugation to exert forces that partially or fully mimic Earth-level gravity in space is a possible countermeasure to mitigate the effects of microgravity on the eye. Therefore, we subjected mice on the International Space Station (ISS) to microgravity (0 G) or artificial gravity by centrifugation at 0.33 G, 0.67 G, and 1 G, and then performed RNA sequencing (RNA-seq) on optic nerve and retinal tissue after returning them to Earth alive. We find that the microgravity environment induces transcriptomic changes in the optic nerve and retina consistent with an increased oxidative stress load, inflammation, apoptosis, and lipid metabolic stress. We also find that adding artificial gravity on board the ISS attenuates the transcriptomic response to microgravity in a dose-dependent manner. Such attenuation may effectively protect from and mitigate spaceflight-induced detrimental effects on ocular tissue.

## 1. Introduction

Physiological and pathological neuro-ophthalmic abnormalities have been observed in astronauts that have undergone prolonged spaceflight, a phenomenon known as spaceflight-associated neuro-ocular syndrome (SANS) [1,2,3]. Our previous rodent flight studies have shown that spaceflight induces adverse effects on retinal structure, blood–retinal barrier (BRB) integrity, and retinal function [4,5]. More recent research also reports optic nerve swelling in astronauts and pathological changes around the optic nerve disc [6]. These findings warrant research on the role of spaceflight conditions, including microgravity (MG), on the optic nerve (ON) and retina (RTN) in the development of SANS [7]. To date, few studies have investigated the mechanisms and factors that contribute to the development of SANS.

Changes in gravity are critical aspects of spaceflight which may contribute to adverse responses. Artificial gravity (AG) by centrifugal force is a potential countermeasure to mitigate the effects of MG on board the International Space Station (ISS) [8]. Rodent centrifugation studies on board the ISS provide opportunities to assess the effectiveness of AG in space. Prior work from the Japan Aerospace Exploration Agency (JAXA) mouse habitat unit-1 (MHU-1) showed that exposure to 1 G of AG effectively reduced endothelial cell damage and increased cellular organization and function in the RTN compared to the MG group without centrifugation [9]. Results from the same flight study also showed that mice exposed to artificial 1 G had reduced musculoskeletal deterioration associated with spaceflight [10]. These studies provide evidence that AG may protect against or mitigate the effects of MG. However, it is currently not known what gravity level and duration of gravitational force must be maintained to prevent or reduce retinal damage.

For the current study, AG was generated by onboard short-radius centrifuges installed in the Centrifuge-Equipped Biological Experiment Facility (CBEF) on board the ISS, created by JAXA [10]. Mice were exposed to either MG alone (0 G), or to 0.33 G, 0.67 G, or 1 G by centrifugation for the duration of a 30-day flight mission. Our in-depth transcriptomic analysis allowed us to robustly characterize and compare molecular responses in the ON and RTN. This study aims to evaluate whether AG could mitigate the deleterious effects of MG on the RTN and ON during spaceflight. Our study also aims to compare the effectiveness of different AG levels and define the relationship between gravitational dose and cellular response by assessing an oxidative stress biomarker and transcriptomic endpoints. This information is expected to provide valuable knowledge of the mechanisms that contribute to ON/RTN dysfunction and facilitate the development of an optimal AG protocol for eye protection during spaceflight.

## 2. Results

### 2.1. RNA-Seq of the Mouse RTN and ON During Spaceflight with Progressive AG

We performed RNA-seq of the ON and RTN of MHU-8 mice under MG, as well as under 0.33 G, 0.67 G, and 1 G of AG, during spaceflight for the flight groups (FLT) as well as for habitat ground control (HGC) mice, which were maintained in housing conditions that were identical to those of the FLT groups. For each FLT group, 5–6 biological replicates were used, and 12 biological replicates were used for the HGC group. MultiQC reports [11] showed that all samples had uniformly high sequencing quality across both read pairs (Appendix A). Normalized gene expression was strongly correlated between replicates of the same experimental group (Appendix A).

To verify cell-type-specific enrichment within our samples, we looked at the gene expression of specific marker genes known to be enriched in either the ON or RTN relative to the other. *Ptx3* and *Mbp* showed high RNA-seq expression in all ON treatment groups and low expression in all RTN treatment groups (Appendix A), consistent with the enrichment of these genes observed by quantitative Polymerase Chain Reaction (*Ptx3*) and RNA-seq (*Ptx3* and *Mbp*) in the ON relative to the RTN [12]. Similarly, *Rho* and *Pou4f1* showed high RNA-seq expression in all RTN treatment groups and low expression in all ON treatment groups (Appendix A); *Rho* was previously shown by RNA-seq to be enriched in the RTN and depleted in the ON [12]/while *Pou4f1* (*Brn3a*) is a gold-standard retinal ganglion cell marker in mice [13].

### 2.2. Spaceflight Induces Robust Transcriptomic Responses in the Mouse ON and RTN

We first compared FLT 0 G vs. HGC RNA-seq in order to determine the effects of spaceflight on transcription in the ON and RTN. We identified a total of 206 significantly differentially expressed genes (DEGs) in the ON (Figure 1a), and 285 DEGs in the RTN (Figure 1b). *Scgb2b12*, *Bpifb9b*, *Bpifb9a*, *Pnmt*, and *Gm10408* were the top five DEGs by magnitude in the ON (Figure 1c), while *Muc5b*, *Klf17*, *Gm9139*, *Scgb1c1*, and *Serpinb6d* were the top five in the RTN (Figure 1d). The majority of DEGs in the ON were distinct from those in the RTN, with only 28 DEGs being common to both the ON and RTN (Figure 1e). These findings demonstrate substantial transcriptional changes in response to spaceflight conditions in both the ON and RTN and warrant further investigation into their functional roles in MG adaptation and the development of SANS.

Next, we conducted a pathway analysis of the significant FLT 0 G vs. HGC DEGs. Forkhead box O (FoxO) signaling, PI3k-Akt signaling, and p53 signaling were among the top pathways by significance in the ON (Figure 2a), whereas peroxisome proliferator-activated receptor (PPAR) signaling and p53 signaling were among the top pathways in the RTN (Figure 2b). Interestingly, p53 signaling was significantly enriched in the ON both after optic nerve crush and in a model of glaucoma [12], suggesting some overlap in the transcriptomic responses to spaceflight, optic nerve crush, and glaucoma. Indeed, we identified a subset of FLT 0 G vs. HGC DEGs that are also differentially expressed in published optic nerve crush and/or glaucoma models [12] in each of the ON and RTN (Appendix A).

### 2.3. AG Attenuates the Transcriptomic Response to Spaceflight in the ON and RTN

Next, we compared spaceflight gene expression under progressive amounts of AG and found that the numbers of DEGs relative to the HGC group trended downward when going from 0 G to 1 G of AG in both the ON and RTN (Appendix A). A major exception to this trend was in the ON at 0.33 G, which had a larger number of DEGs than all other treatment groups (Appendix A). We also looked specifically at the effect of AG on the normalized expression of the FLT 0 G vs. HGC DEGs, as well as global gene expression patterns. In both cases, as AG progressed toward 1 G, gene expression tended to revert to the pattern of the HGC group (Figure 3a–f and Appendix A). The ON at 0.33 G was again an outlier of this pattern. However, in a principal component analysis (PCA) plot of individual replicates, it can be seen that three replicates of the ON at 0.33 G were extreme outliers that did not cluster with each other or with the other three 0.33 G replicates, whereas the other three replicates clustered together and followed the trend of progressively reverting to the gene expression pattern of the HGC group when moving from 0 G to 1 G of AG (Appendix A). It is therefore likely that the anomalous behavior of the ON at 0.33 G is due to some unidentified technical or experimental artifacts in three of the replicates, and not reflective of an actual biological response. Taken together, these results demonstrate that AG attenuates gene expression changes induced by spaceflight in both the ON and RTN in a dose-dependent manner, with the strongest attenuation occurring at 1 G.

We then looked at the effect of AG on genes within select pathways among those most significantly altered by spaceflight (Figure 2a,b). The expression of a subset of genes within the FoxO signaling pathway and p53 signaling pathway in the ON, and the PPAR signaling pathway and p53 signaling pathway in the RTN, showed a trend of progressive attenuation as AG was increased from 0 G to 1 G (Figure 4a–d). This is also true of the PI3k-Akt signaling pathway in the ON (Appendix A). Our results therefore suggest that not only are these pathways potentially important in the transcriptomic response to spaceflight within the ON/RTN, but that AG might also attenuate the alteration of these pathways during spaceflight.

To obtain further evidence of AG-attenuated transcriptional responses during spaceflight, we compared the FLT 0 G vs. HGC DEGs to the FLT 0 G vs. FLT 1 G DEGs. If robust attenuation of the spaceflight response occurs at 1 G, then the transcriptome at 1 G should be similar to that of the HGC group, and so the FLT 0 G vs. HGC DEGs should tend to change in the same direction as the FLT 0 G vs. FLT 1 G DEGs. Indeed, significantly more DEGs changed in the same direction than in the opposite direction in both the ON and RTN (Figure 5a,b), and, in fact, no genes changed in the opposite direction in either the ON or RTN. This supports the idea that adding 1 G of AG on the ISS attenuates the effects of MG. Mice at 1 G of AG maintained similar levels of transcriptome expression to HGC mice, as exemplified by *Pdk4* in the ON (Figure 5c) and *Cdkn1a* in the RTN (Figure 5d).

### 2.4. AG Attenuates a Spaceflight-Induced Increase in the Oxidative Stress Biomarker 4-Hydroxynonenal (4-HNE) in the RTN

Given the prevalence of oxidative stress-related pathways in our spaceflight data, we measured the effects of spaceflight and AG on the expression of 4-HNE in the RTN by immunofluorescence. 4-HNE is considered an oxidative stress biomarker for lipid peroxidation and plays an important role in oxidative stress-mediated responses [14,15]. Enhanced 4-HNE staining was seen in the ganglion cell layer (GCL) of the 0 G and 0.33 G groups compared to all other groups (Figure 6a). Spaceflight significantly increased the immunoreactivity of 4-HNE relative to that observed in HGC animals (*p* < 0.05), while 0.67 G and 1 G of AG rendered 4-HNE immunoreactivity indistinguishable from that observed in HGC animals (Figure 6b). These results demonstrate a spaceflight-induced increase in oxidative stress in the RTN, which can be attenuated by at least 0.67 G of AG.

## 3. Discussion

The current study demonstrates that spaceflight induces transcriptomic changes in optic nerve and retinal tissue. Our pathway analysis suggests that these changes may involve oxidative stress, inflammation, apoptosis, and lipid metabolic stress. Such pathway changes could lead to ocular injury and degeneration. Our study is the first to assess the role of different doses of AG for the duration of spaceflight on the ISS as a countermeasure to protect and mitigate the effects of MG on the eye. In addition, our study is the first to observe spaceflight-induced transcriptomic responses specifically within the ON and RTN. The majority of our data documented that continuous AG on the ISS attenuates the transcriptomic response to spaceflight in a dose-dependent manner, with 1 G providing the strongest attenuation. These results suggest that added AG can be implemented to prevent or reduce the ocular physiological and cellular damage induced by spaceflight stressors. As a potential preventive approach, optimal centrifugation prescriptions may modulate the normal tissue response to spaceflight conditions or increase the tolerance of the retina and optic nerves to injuries.

In this study, the p53 signaling pathway is among the top pathways enriched by pathway analysis in the ON and RTN. Oxidative stress can activate a variety of transcription factors, including p53 [16]. Dysregulation of p53 pathways then leads to impaired DNA repair, cell cycle arrest, and increased apoptosis [17]. Recent research has also shown that p53 plays an important role in the progression of retinal degenerative disorders including age-related macular degeneration [18]. Our results reveal that the alteration of genes within the p53 signaling pathway was progressively attenuated by AG from 0.33 G to 1 G. Oxidative stress has been strongly implicated in spaceflight and in space radiation-induced ocular damage [19,20]. The measurement of the oxidative stress biomarker 4-hydroxynonenal in the RTN in this study confirms both spaceflight-induced oxidative stress within the RTN, as well as attenuation of oxidative stress within the RTN at 0.67 G and 1 G of AG compared to 0 G. By modulating the level of oxidative stress with added AG, it may be possible to mitigate the damaging effects of MG on ocular tissue.

The FoxO signaling pathway is significantly enriched by pathway analysis in the ON due to spaceflight. Furthermore, the expression of genes within this pathway shows a pattern of progressive attenuation in moving from 0 G to 1 G of AG. FoxO proteins act as key regulators in a wide range of crucial cellular processes involving stress, metabolism, cell cycle arrest, and apoptosis [21]. FoxO signaling dysregulation due to spaceflight conditions may lead to a negative impact on ON health.

Similarly, the PI3k-Akt signaling pathway is significantly enriched by spaceflight in the ON, and patterns of gene expression within this pathway are progressively attenuated by increasing AG. The PI3k-Akt signaling pathway plays a key regulatory role in cellular processes, including apoptosis, cell survival, cell cycle progression, and metabolism [22]. Its influence in mediating neurological effects has been documented [23]. There is a strong link between activation of the PI3k-Akt signaling pathway and oxidative stress/inflammation [24,25]. Studies have shown that the PI3K-Akt signaling pathway is involved in the regulation of oxidative stress-induced apoptosis [26]. Interestingly, the PI3k-Akt signaling pathway is documented to be the most important pathway interacting with FoxO in regulating different physiological processes and pathological events or diseases [27].

In the RTN, the PPAR signaling pathway is significantly enriched due to spaceflight by pathway analysis, and genes within this pathway are progressively attenuated by increased AG doses. Genes within the PPAR signaling pathway play crucial roles in glucose and lipid metabolism, oxidative stress, and chronic inflammation [28]. A simulated MG study by hindlimb suspension revealed the differential expression of a number of transcripts, including the PPARs that regulate lipid and glucose metabolism in the muscle [29].

Our data from this study indicate that adding AG for the duration of spaceflight on the ISS, especially higher doses of 0.67 G and 1 G, can bring gene expression patterns closer to those of the grounded control group. This study provides evidence that in-flight AG can mitigate some of the effects of weightlessness during spaceflight. In this study, we found that the ON and RTN exhibited mostly distinct transcriptomic responses. The cellular components and structure of the ON and RTN are notably different in terms of cell types, astrocytic features, vascular density, and metabolic activity, as well as permeability and blood barrier properties [12]. Microvessels in the prelaminar region of the ON head lack blood–brain barrier (BBB) characteristic proteins, while the RTN has BBB biomarkers [30]. More recently, a study found that notable changes in the expression of several biomarkers related to the blood barrier in the ON head following space radiation exposure differ from those of the RTN [20]. The largely unique gene expression profiles observed in this study in the ON and RTN point to a complex response to space-specific environmental insults. Further studies are needed to investigate the underlying mechanisms linking the observed transcriptomic changes to the development of pathologic neuro-ophthalmic abnormalities.

Spaceflight stressors besides MG, including cosmic radiation, magnetic changes, hypergravity during transport, and other space-specific environmental influences, might all have effects on ocular health [31]. This study did not explicitly control for additional potential stressors, including hypergravity, cosmic radiation, and magnetic changes, in the HGC mice. Although MG and cosmic radiation are considered the primary sources of harmful effects during spaceflight [31], secondary sources such as hypergravity may have made minor contributions to the observed transcriptomic changes during spaceflight. Further, during a short, 30-day mission, the mice are only expected to be exposed to relatively low doses of cosmic radiation. However, there is evidence that MG and cosmic radiation may act synergistically [31]. The fact that there are relatively few DEGs and low 4-HNE staining at 0.67–1 G (Appendix A and Figure 6b, respectively) may therefore suggest either that spaceflight stressors other than MG did not have a major effect at all in this study, or that they did have a substantial effect synergistically with MG that was attenuated under substantial AG. Stressors like cosmic radiation are likely to be more substantial during spaceflight of a longer duration. It is possible that continuous AG above 0.67 G may help protect against the effects of MG and other spaceflight stressors during longer missions on the ISS, though this is speculative. It will be important in future studies to expose mice to a longer duration of spaceflight on the ISS to determine the effectiveness of AG in attenuating the effects of spaceflight stressors.

In order for the collected data from the current study to be comparable to the data from past MHU missions [9,32,33], only male mice were used in this study. However, we recognize the importance of studying both sexes for space research to understand how sex and gender differences impact long-term health in astronauts. It will be important in future research to incorporate both male and female mice to better understand their differences in response to space-specific environmental stressors.

In the present study, we observed increased 4-HNE activity in the retinal GCL of the FLT 0 G and 0.33 G groups. The exact cell types of the positive staining need to be further characterized by double staining with specific biomarkers. However, the morphology of these 4-HNE signals resembles Müller cell endfeet. Müller cells are glial cells in the retina that provide metabolic and homeostatic support to neurons and photoreceptors. Increased activation and hypertrophy of Müller cells occurs in a process called gliosis in response to retinal damage or stress [32,33]. The effect of retinal gliosis on neurodegeneration has been reported [34]. Reduced 4-HNE staining on Müller cells by higher doses of AG may indicate protective effects against the impact of MG on the retina by mechanisms of attenuating oxidative stress and altering Müller glial activation and gliosis processes.

## 4. Materials and Methods

### 4.1. Mouse Habitat Unit-8 and Mouse Groups

A total of 24 male C57BL/6J mice (Jackson laboratories, Inc., Bar Harbor, ME, USA) were individually housed on Earth during a four-week period of acclimation to housing, food, and water conditions. The mice were then transferred to the Transportation Cage Unit (TCU) and launched at ~12 weeks of age from the Kennedy Space Center (KSC) to the ISS on SpaceX CRS-27 (Dragon) on 15 March 2023 (GMT) for a 30-day MHU-8 mission. Within 24 h after arriving at the ISS, the animals were individually housed in the JAXA Habitat Cage Units (HCUs) located in the Kibo centrifuge hardware Cell Biology Experimental Facility on the ISS. The 24 flight mice were subdivided into four groups. The first group of flight mice were exposed to ambient microgravity conditions (0 G group), the second and third groups of flight mice were exposed to continuous AG of 0.33 G and 0.67 G, respectively, and the fourth group was exposed to continuous AG of 1 G. AG was achieved through the use of short-arm centrifuges for the duration of the MHU-8 mission, with only brief stoppages for husbandry and experimental procedures several times throughout the mission. The AG mice were subjected to AG immediately after being transferred to their HCUs. The rotation rates for generating 0.33 G, 0.67 G, and 1 G were 44 rpm, 63 rpm, and 77 rpm, respectively. The flight mice then splashed down in the Atlantic Ocean on 15 April 2023, and returned to Earth alive. A total of 12 male C57BL/6J HGC mice from the same cohort as the FLT mice were also housed individually during a four-week period of acclimation and were then maintained in similar habitat conditions as the FLT mice, inside the ISSES Chambers, with operations conducted with a 3-day delay relative to the FLT mice. All FLT and HGC mice received the same ad libitum access to food and water. Within 6 h of landing, all FLT mice were euthanized by inhalation of 3% isoflurane and then maintained at 1.5–2% for blood collection following euthanasia, and their eyes were removed and prepared for analysis. HGC mice were euthanized 3 days later by the same method.

Animal experiments were approved by the Institutional Animal Care and Use Committee (IACUC) of JAXA and NASA. No protocol was required from the Loma Linda University (LLU) IACUC, since only archived frozen and fixed tissues were obtained after euthanasia. The research with vertebrate animals was conducted in strict accordance with guidelines and applicable laws in Japan and the United States of America.

### 4.2. Mouse Eye Tissue Harvest and Preservation

After necropsy, mouse eyes were harvested. The left eyes from each experimental group were fixed in 4% paraformaldehyde in phosphate-buffered saline (PBS) for immunohistochemistry (IHC) assays. The retina and optic nerve of the right eyes were dissected out from the eye globe [35] and placed individually in sterile cryovials, snap frozen in liquid nitrogen, and kept at −80 °C prior to RNA extraction and RNA-seq.

### 4.3. Retina RNA Extraction

RNA was extracted using the AllPrep DNA/RNA Mini Kit (Qiagen, Valencia, CA, USA) following the GeneLab SOP 3.1-Qiagen AllPrep DNA/RNA Mini with Qiagen RNase-Free DNase set (https://github.com/nasa/GeneLab-sampleProcessing/tree/master/SOP_pdfs (accessed on 27 June 2023)). A homogenization buffer for RNA purification was made by adding 1:100 beta-mercaptoethanol to Buffer RLT (Qiagen, Valencia, CA, USA) and was kept on ice until use. MHU-8 retina tissues processed in this experiment were not weighted before RNA extraction to avoid thawing, and 900 μL of cold homogenization buffer was added directly into the tissue tubes. Homogenization was performed using the rotor stator homogenizer (Polytron, Raymertown, NY, USA), with 3 repetitions of 10 s each at a speed of 21,000 as per the SOP 2.2-Tissue homogenization using Polytron Rotor Stator Homogenizer (https://github.com/nasa/GeneLab-sampleProcessing/tree/master/SOP_pdfs (accessed on 27 June 2023)). Homogenates were centrifuged for 5 min at room temperature and 14,000 G to remove cell debris. The supernatant from each sample was then used to isolate and purify RNA following the manufacturer’s protocol. RNA was eluted in 40 μL RNAse-free H_2_O. Concentrations for all RNA samples were measured using the Qubit 3.0 Fluorometer (Thermo Fisher Scientific, Waltham, MA, USA) with the Qubit RNA BR kit following the SOP 4.1-RNA/DNA/miRNA/cDNA Quantification using Qubit Fluorimeter (https://github.com/nasa/GeneLab-sampleProcessing/tree/master/SOP_pdfs (accessed on 27 June 2023)). RNA quality was assessed using the Agilent 4200 TapeStation with the Agilent RNA ScreenTape (Agilent Technologies, Santa Clara, CA, USA).

### 4.4. Optic Nerve RNA Extraction

RNA was extracted using the RNeasy Universal kit (Qiagen, Valencia, CA, USA) following the manufacturer’s protocol. QIAzol was used as a homogenization buffer and was kept on ice until use. MHU-8 optic nerve tissues processed in this experiment were not weighted before RNA extraction to avoid thawing, and cold QIAzol homogenization buffer was added directly into the tissue tubes. Homogenization was performed using the rotor stator homogenizer (Polytron, Raymertown, NY, USA), with 3 repetitions of 10 s each at speed of 21,000 as per the SOP 2.2-Tissue homogenization using Polytron Rotor Stator Homogenizer (https://github.com/nasa/GeneLab-sampleProcessing/tree/master/SOP_pdfs (accessed on 27 June 2023)). Homogenates were centrifuged for 5 min at room temperature and 14,000 G to remove cell debris. The supernatant from each sample was then used to isolate and purify RNA following the manufacturer’s protocol. RNA was eluted in 40 μL RNAse-free H_2_O. Concentrations for all RNA samples were measured using the Qubit 3.0 Fluorometer (Thermo Fisher Scientific, Waltham, MA, USA) with Qubit RNA BR or HS kits following the SOP 4.1-RNA/DNA/miRNA/cDNA Quantification using Qubit Fluorimeter (https://github.com/nasa/GeneLab-sampleProcessing/tree/master/SOP_pdfs (accessed on 27 June 2023)). RNA quality was assessed using the Agilent 4200 TapeStation with the Agilent RNA ScreenTape (Agilent Technologies, Santa Clara, CA, USA).

### 4.5. Sequencing Library Construction

ERCC ExFold RNA Spike-In Mixes (Thermo Fisher Scientific, Waltham, MA, USA, Cat 4456739, v92), Mix 1 or Mix 2, were added on the day of library prep at the concentrations suggested by the manufacturer’s protocol. Spike-in data were used for sample quality control. RNA ribo depletion and library preparation were conducted with Illumina Stranded Total RNA Prep with Ribo-Zero Plus (Illumina Inc., San Diego, CA, USA). Input RNA was approximately 100 ng with RIN > 4. For RTN samples FLT17 and FLT02, and ON samples FLT11, FLT21, FLT04, FLT23, FLT12, FLT02, and HGC05, RNA input was below 30 ng, and ERCC spike-in and quality control could not be performed for these samples. Unique dual index adapters (Illumina Inc., San Diego, CA, USA) were used, and 12 cycles of PCR were performed using the BioRad C1000 Touch (BIO RAD, Hercules, CA, USA). Library fragment size was assessed using Agilent 4200 TapeStation with D1000 DNA ScreenTape (Agilent Technologies, Santa Clara, CA, USA) following the GeneLab SOP 6.3 Quality analysis of sequencing libraries using 4200 TapeStation System with D1000 reagent kit (https://github.com/nasa/GeneLab-sampleProcessing/tree/master/SOP_pdfs (accessed on 27 June 2023)). Pooled library concentration was measured by the Qubit 4 Fluorometer (ThermoFisher Scentific, Waltham, MA, USA) following the SOP 4.1-RNA/DNA/miRNA/cDNA Quantification using Qubit Fluorimeter (https://github.com/nasa/GeneLab-sampleProcessing/tree/master/SOP_pdfs (accessed on 27 June 2023)). Library quality assessment was performed on iSeq100 (Illumina, San Diego, CA, USA).

### 4.6. Sequencing

RNA sequencing was performed by the GeneLab Sample Processing Lab on Illumina NovaSeq 6000. Sequencing was set up as follows: Read 1: 151 bp; Index1: 17 bp; Index2: 8 bp; Read 2: 151 bp. PhiX Sequencing Control V3 (Illumina Inc., San Diego, CA, USA) was included as a control and to increase library diversity. The protocol followed is the SOP 7.1-Setting up NovaSeq 6000 and iSeq 100 Sequencers (https://github.com/nasa/GeneLab-sampleProcessing/tree/master/SOP_pdfs (accessed on 27 June 2023)).

### 4.7. RNA-Seq Data Analysis

Raw fastq files were assessed for percent rRNA using HTStream SeqScreener (version 1.3.2) [36], and adaptors and low-quality bases were trimmed using Trim Galore! (version 0.6.7) [37] powered by Cutadapt (version 3.7) [38]. Trimmed fastq file quality was then evaluated with FastQC (version 0.11.9) [39], and MultiQC (version 1.1.9) [11] was used to generate MultiQC reports. Trimmed fastq files were then mapped with STAR (version 2.7.10a) [40]. We used the Ensembl [41] release 107, genome version GRCm39 (Mus_musculus.GRCm39.dna.primary_assembly.fa) concatenated with ERCC92.fa from ThermoFisher (ThermoFisher Scentific, Waltham, MA, USA) as a reference genome. The following gtf annotation file was used: Mus_musculus.GRCm39.107.gtf concatenated with the ERCC92.gtf file from ThermoFisher.

Aligned reads were assessed for strandedness using RSeQC Infer Experiment (version 4.0.0) [42,43]; then, aligned reads from all samples were quantified using RSEM (version 1.3.1) [44], with strandedness set to reverse. Quantification data were imported to R (version 4.1.3) with tximport (version 1.22.0) [45] and normalized with DESeq2 (version 1.34.0) [46]; all groups were compared pairwise using the Wald test, and the likelihood ratio test was used to generate the *F* statistic *p*-value. Gene annotations were assigned using the following Bioconductor and annotation packages: STRINGdb (v2.8.4) [47,48,49], PANTHER.db (v1.0.11) [50,51,52], and org.Mm.eg.db (v3.15.0). Genes with |log2 FC| > 1 and Adjusted *p* < 0.05 were selected as significant DEGs.

PCA plots for the ON and RTN were made separately using the normalized gene expression data obtained from DESeq2. The analysis was done in R (version 4.0.3). The R function aov() [53] was used to perform a one-way analysis of variance (ANOVA) on each gene using each treatment as a factor, and genes with *p* < 0.05 were selected for input into the PCA. PCAs were performed using the R function PCA() [54,55,56].

Pathway analysis and plotting were performed separately on the ON and RTN DEGs (FLT 0 G vs. HGC) using R (version 4.0.3) with the enrichR package (version 3.2) [57,58]. setEnrichrSite(“Enrichr”) was used for performing general enrichment analysis. Only the KEGG_2019_Mouse [59] library was used for analysis.

### 4.8. Comparison of DEGs Overlapping Between Spaceflight, Glaucoma, and Optic Nerve Crush Models

Published glaucoma and optic nerve crush DEG tables (both relative to healthy controls) were obtained [12], and genes with |log2 FC| > 1 and Adjusted *p* < 0.05 were taken to be significant. Separate lists were obtained for the ON and RTN, which contained DEGs at all time points observed in the study for both optic nerve crush and glaucoma models. Optic nerve crush and glaucoma DEGs in unmyelinated optic nerve (UON) and myelinated optic nerve (MON) were combined into a single list for the ON. We then determined the overlaps between spaceflight (FLT 0 G vs. HGC) DEGs in the ON and the optic nerve crush + glaucoma DEGs in the ON, and similarly determined the overlaps between spaceflight DEGs in the RTN and the optic nerve crush + glaucoma DEGs in RTN.

### 4.9. Statistical Comparison of FLT 0 G vs. HGC and FLT 0 G vs. FLT 1 G DEG Overlaps

For the ON and RTN separately, we counted the number of significant DEGs that changed in the same direction in the two comparisons (FLT 0 G vs. HGC and FLT 0 G vs. FLT 1 G) according to the log2 FC, and we similarly counted the number of DEGs that changed in the opposite direction. Let S = the number of DEGs changing in the same direction, O = the number of DEGs changing in the opposite direction, and n = S + O. We thus performed a binomial test using the function “binom.test” in R version 4.0.3, with x = S, n = N, and *p* = 0.5.

### 4.10. Immunostaining for Oxidative Stress Biomarker Evaluation

Paraffin-embedded sections (six µm) of the left eye, roughly 100 mm apart, were used for analysis (n = 5–6/group). To evaluate oxidative damage in the retina, immunostaining was performed on ocular sections using the 4-HNE antibody specific to lipid peroxidation (catalog no. HNE11-S, Alpha Diagnostic International Inc., San Antonio, TX, USA). Sections were incubated with the anti-4-HNE antibody at 4 °C for 2 h followed by a donkey anti-rabbit IgG fluorescence-conjugated secondary antibody (catalog no. A21206, Invitrogen Corp., Waltham, MA, USA) for 2 h at room temperature and counterstained with DAPI.

Six to ten field images were captured with a BZ-X700 inverted fluorescence microscope (Keyence Corp., Itasca, IL, USA) at 20× magnification spanning the entire retina sections. To determine 4-HNE immunoreactivity, fluorescence intensity (red channel for 4-HNE) was measured in these captured images of each section and calculated using ImageJ software (V1.53) [60]. The data were then extracted and averaged across 5 retina sections per eye within the group.

## 5. Conclusions

We conclude that adding artificial gravity by centrifugation on board the ISS, especially near 1 G, can significantly reduce transcriptomic alteration induced by spaceflight in the optic nerve and retina. Such attenuation may effectively mitigate spaceflight-induced detrimental effects on ocular tissue. Clearly, more research on the ocular response to microgravity and its interaction with other spaceflight physiological stressors is warranted. Further investigation will also be needed to extend the observation time after the animals return alive from the centrifugation study on the ISS.

## Figures and Tables

**Figure 1 ijms-25-12041-f001:**
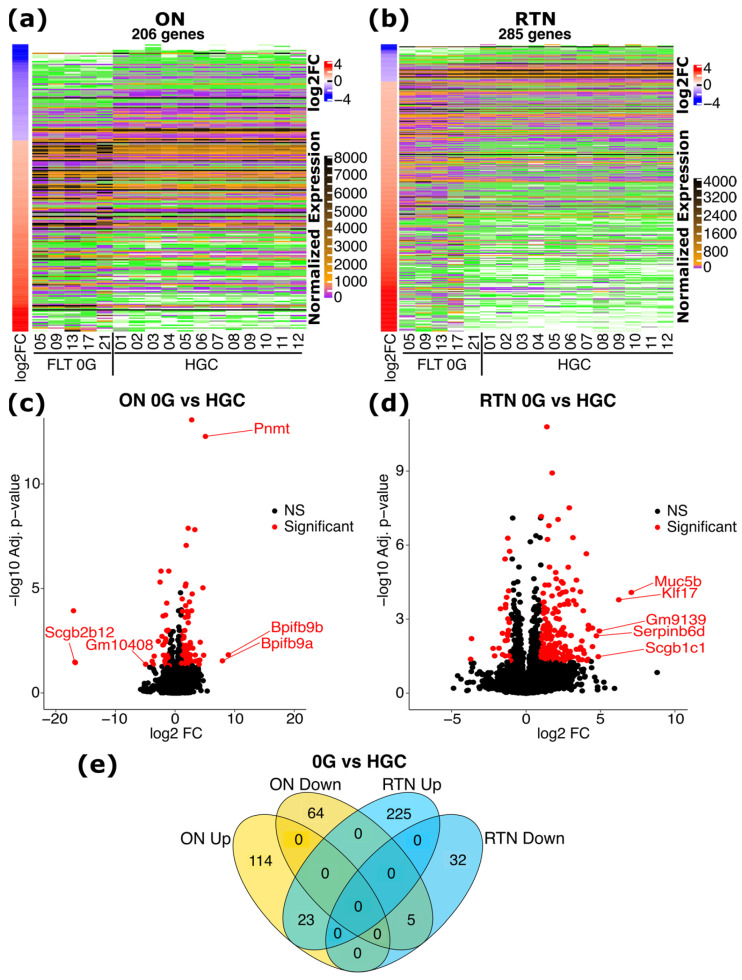
Differentially expressed genes (DEGs) in response to spaceflight in the optic nerve (ON) and retina (RTN). (**a**,**b**) Heatmaps depicting the normalized expression and log2 Fold Change (FC) of the significant DEGs in the spaceflight group (FLT) under 0 G compared to habitat ground control (HGC) mice in the ON and RTN, as indicated. Genes are ordered by |log2 FC|. (**c**,**d**) Volcano plots of the comparison of FLT 0 G vs. HGC in the ON and RTN, as indicated. Significant DEGs are shown in red. For the top 5 DEGs by magnitude of log2 FC that have a unique gene symbol, the gene symbols are displayed. (**e**) Venn diagram showing the number of spaceflight DEGs distinct and overlapping between ON and RTN. DEGs that are upregulated and downregulated at 0 G relative to the HGC group are indicated.

**Figure 2 ijms-25-12041-f002:**
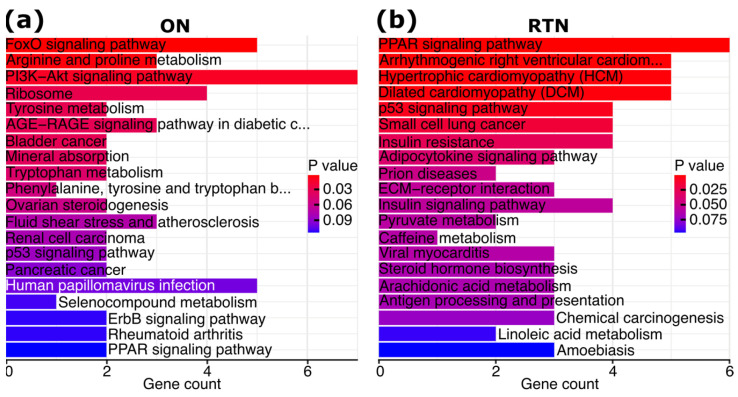
Pathway analysis of spaceflight (FLT 0 G vs. HGC) DEGs. (**a**,**b**) Top 20 most significant pathways enriched among spaceflight DEGs, ordered by significance. The number of spaceflight DEGs within each pathway is shown on the *x*-axis.

**Figure 3 ijms-25-12041-f003:**
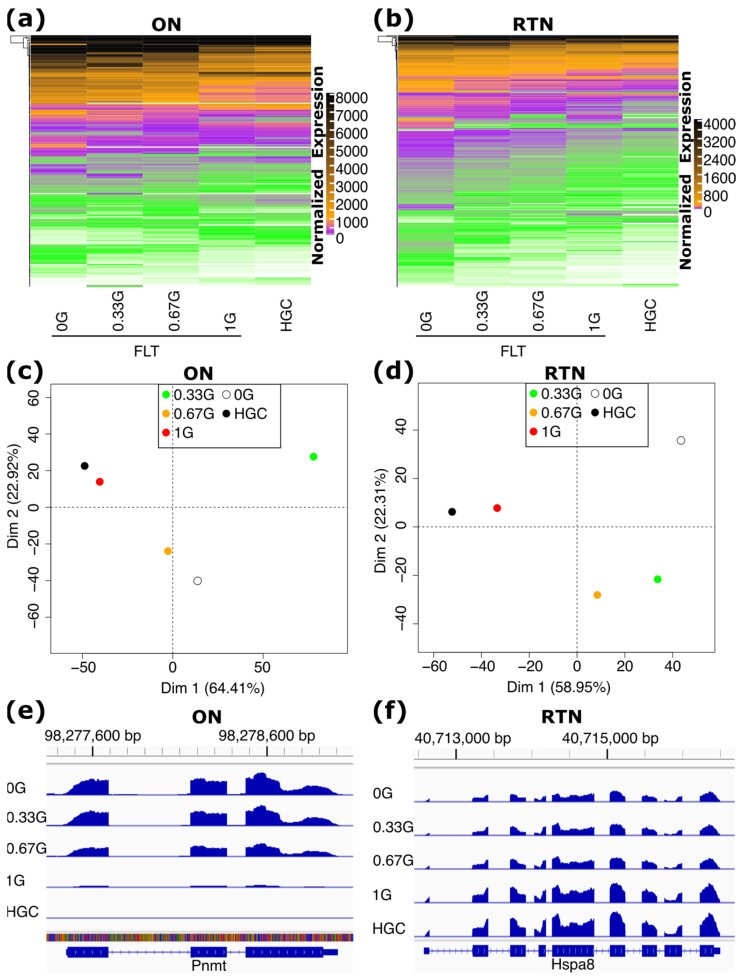
Gene expression during spaceflight under artificial gravity (AG). (**a**,**b**) Heatmaps of the average expression of significant DEGs (FLT 0 G vs. HGC) in the indicated treatment group. Genes are sorted by hierarchical clustering. (**c**,**d**) Principal component analysis (PCA) plots of normalized gene expression data averaged over replicates of each treatment group indicated. (**e**,**f**) Reads per million (RPM)-normalized gene expression of example genes, with replicates of each treatment group pooled.

**Figure 4 ijms-25-12041-f004:**
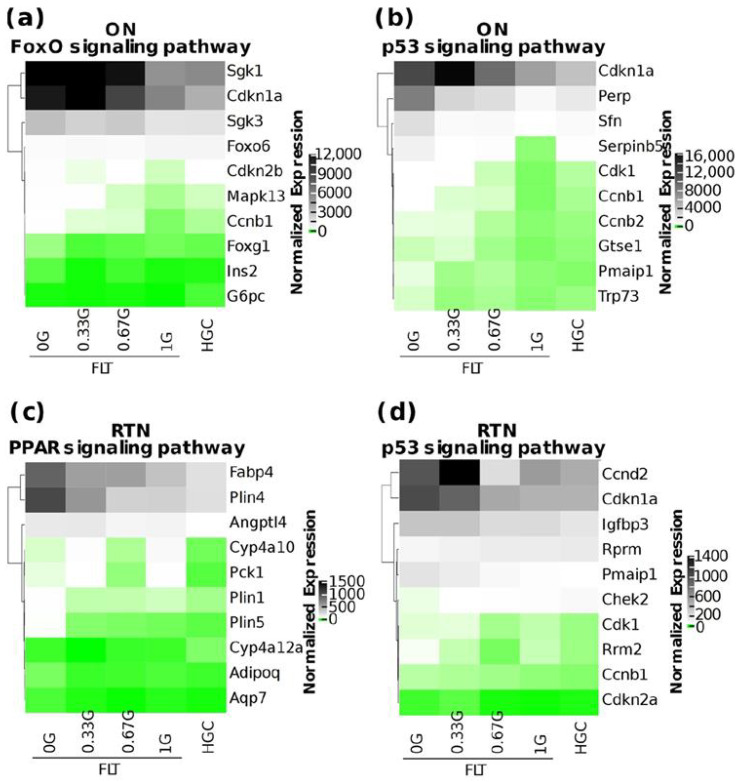
Effect of AG on gene expression in select pathways. (**a**–**d**) Heatmaps of the normalized gene expression, averaged over replicates of the indicated samples and treatment groups, of the genes with the top ten magnitudes of log2 FC in FLT 0 G vs. HGC from the indicated pathways. Gene symbols are shown to the right of each heatmap. Genes are sorted by hierarchical clustering.

**Figure 5 ijms-25-12041-f005:**
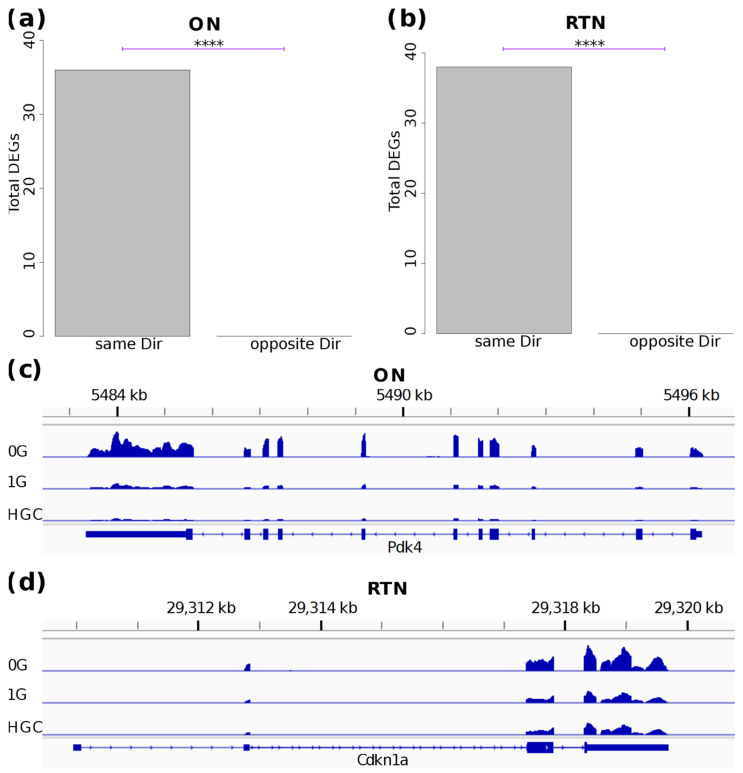
1 G of AG results in a transcriptome that is similar to that of the HGC group. (**a**,**b**) Barplots comparing the FLT 0 G vs. HGC DEGs to the FLT 0 G vs. FLT 1 G DEGs. The number of DEGs changing in the same and opposite direction (Dir) are displayed. *p*-Values were calculated by the binomial test. **** *p* < 10^−10^. (**c**,**d**) RPM-normalized gene expression of example genes, with replicates of each treatment group pooled.

**Figure 6 ijms-25-12041-f006:**
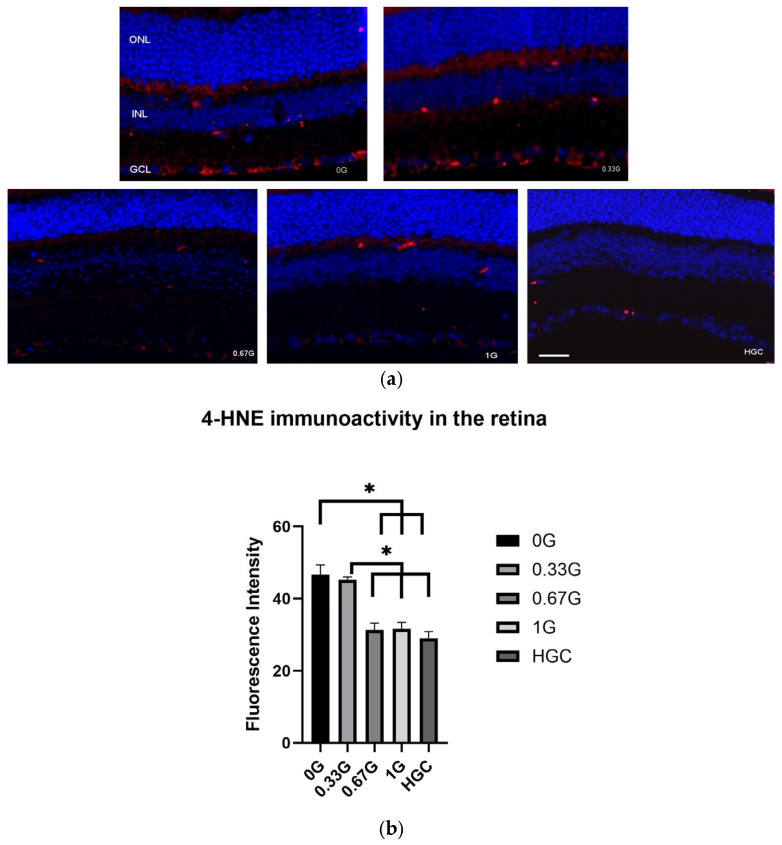
Cellular oxidative damage in the male mouse retina under AG at 0 G, 0.33 G, 0.67 G, and 1 G. (**a**) Representative micrographs of ocular sections from the HGC or spaceflight mice groups for lipid peroxidation by immunostaining with anti-4-hydroxynonenal (4-HNE) antibody are shown. 4-HNE-positive cells were identified with red fluorescence. The nuclei were counterstained with DAPI (blue) in the outer nuclear layer (ONL), inner nuclear layer (INL), and ganglion cell layer (GCL). Scale bar = 50 μm. (**b**) Average fluorescence intensity values for 4-HNE in the retina, represented as mean density ± SEM for 5–6 mice/group. * Significantly increased 4-HNE staining (*p* < 0.05), one-way analysis of variance followed by Tukey’s honestly significant differences test.

## Data Availability

The data presented in this study are openly available in the NASA Open Science Data Repository, accession numbers OSD-758 (https://osdr.nasa.gov/bio/repo/data/studies/OSD-758 (accessed on 12 August 2024) and OSD-759 (https://osdr.nasa.gov/bio/repo/data/studies/OSD-759 (accessed on 12 August 2024)). The GeneLab bioinformatics processing pipeline used in this study is available on the github repository (https://github.com/nasa/GeneLab_Data_Processing/blob/master/RNAseq/Pipeline_GL-DPPD-7101_Versions/GL-DPPD-7101-F.md (accessed on 27 June 2023)). Additional custom analysis scripts used in this study are available on the github repository (https://github.com/ikremsky/Scripts-for-Kremsky-et-al-2024 (accessed on 23 August 2024)).

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
