# Peer review of "Artificial Gravity Attenuates the Transcriptomic Response to Spaceflight in the Optic Nerve and Retina"

_ijms, 2024, doi:10.3390/ijms252212041_

Round 1
Reviewer 1 Report
Comments and Suggestions for Authors
Manuscript ID: ijms-3268072
Title: Artificial Gravity Attenuates the Transcriptomic Response to Spaceflight in the Optic Nerve and Retina
Comments:
This study employed the real space environment of the International Space Station (ISS) to conduct transcriptomic analysis aimed at investigating molecular-level changes associated with SANS pathology. Transcriptomic alterations were observed in ON and RTN samples obtained from mice exposed to microgravity. Notably, these changes were attenuated under conditions of artificial gravity (AG).
1. This reviewer recognizes that hypergravity, experienced during the round-trip from Earth to space, is also an important factor, alongside microgravity, that induces significant physiological changes. In this study, since the mice were returned to Earth alive, they were undoubtedly exposed to hypergravity. While the duration of exposure to hypergravity is certainly shorter than the experimental period in the ISS, the extent of its impact on the mice remains uncertain and may introduce bias into the results.
2. What are the authors thought on the effects of space radiation? Given that the experiments were conducted on the ISS for a duration of 30 days, it is possible that the mice, aside from those in the ground control, may have been influenced by exposure to space radiation.
3. The Veen diagram presented in Figure 1(e) should be regenerated to clearly depict both upregulation and downregulation. For guidance, please refer to Figure 2 in Sci Rep. 2017;7:46107 (doi:10.1038/srep46107).
4. As the authors described in the paragraph 2.3, three mice in the ON at 0.33 G were identified as significant outlier. Were all final analyses conducted with these outliers excluded? Additionally, the sample distribution in all groups of RNT appears broader compared to that of the groups of ON (Figure S2 in Supplementary File S1). Can it be determined whether the RNT samples were well clustered according to the AG conditions? What are the author’s thought on the observed differences in sample distribution between the ON and RNT?
5. In Figure 6, please include the 4-HNE staining image as well. This would enhance the reliability of the findings for the readership.
Reviewer 2 Report
Comments and Suggestions for Authors
The manuscript by Kremsky and collaborators characterizes the protective effects of applying artificial gravity on mice exposed to spaceflight. The study focuses on the transcriptomic changes in the retina and optic nerve (ON) of animals under four different gravity conditions (0G, 0.33G, 0.67G, and 1G), comparing them to animals maintained under normal gravity conditions (HGC). The authors report over 200 transcripts that exhibit changes in both the retina and ON, linked to pathways commonly associated with degeneration, apoptosis, and oxidative stress. Notably, they find that transcript levels in animals exposed to 0.67G and 1G closely resemble those in the HGC group, suggesting a potential preventive effect against the development of spaceflight-associated neuro-ocular syndrome (SANS). The manuscript is well-organized, and the methods are described in sufficient detail to ensure reproducibility. The sample sizes for both transcriptomic and anatomical analyses are appropriate, and the statistical analyses are generally sound. However, while the study offers valuable insights into the potential protective role of artificial gravity in the retina and ON, several areas require further consideration. First, the manuscript lacks direct evidence of pathological events beyond the assessment of oxidative stress via 4-HNE, which alone does not directly imply visual deficits. This is particularly relevant for interpreting the study’s results. Including visual behavioral tests, functional assays, or additional anatomical assessments would strengthen the conclusions and provide more comprehensive evidence of protection against SANS-related pathology. The manuscript is generally free of major grammatical errors. Below are some additional points and suggestions for further improvement:
1. Euthanasia. While the authors mention that all animals were treated in accordance with Japanese and American guidelines for animal welfare, the specific method of euthanasia (line 286) is not detailed. Could the authors specify how euthanasia was carried out?
2. I would recommend including images of the 4-HNE staining in Figure 6. This would not only help assess the overall changes in fluorescence intensity across the retina but also indicate the precise location of the fluorescence. Since different retinal populations are located in distinct layers, knowing which part of the retina shows an increase in 4-HNE immunoreactivity may help identify the specific cell types most affected.
3. Have the authors evaluated oxidative stress in the optic nerve (ON) by comparing the different artificial gravity (AG) groups (0G, 0.33G, 0.67G, and 1G) to the HGC group? This could provide additional insights into the protective effects of AG on the ON.
4. Anatomical confirmation. Although the authors acknowledge the need for further anatomical assays in their limitations section, I believe it is important to include additional assessments to support the conclusion that the transcriptomic changes lead to retinal degeneration. This could be achieved by evaluating: i/ cell death markers, such as apoptosis (e.g., caspase-3) or performing a TUNEL assay; or ii/ neuronal cell loss, for instance using markers like RBPMS or Brn3a for retinal ganglion cells, opsins for photoreceptors, or even nuclear staining (e.g., DAPI) to quantify the number of nuclei rows, which could estimate degeneration in any nuclear layer of the retina. Significant differences between the 0G and 1G groups (or HGC) would further support the conclusion.
5. Lack of Functional and Behavioral Assessments: Another notable weakness is the absence of functional or behavioral assessments to link transcriptomic changes to visual deficits or pathological alterations in the optic nerve. While this study focuses on transcriptomic changes in the retina and ON under different AG conditions, it is critical to provide evidence of their visual impact. Behavioral tests, such as visual acuity (qOMR), or functional tests (ERG, VEP), could be performed without requiring animal sacrifice and would offer valuable insight into how these changes affect vision.
6. The authors found that the transcriptomic signatures of AG at 0.67G and 1G (particularly 1G) closely resemble those of the HGC retinas and optic nerves. However, it is important to note that exposure to space can involve other factors, such as cosmic radiation, magnetic changes, and other space-specific environmental influences (PMID: 37374046), which could contribute to the development of SANS. Could the absence of significant changes in 4-HNE staining for the 0.67G and 1G groups suggest that these space-related factors do not significantly contribute to oxidative stress or retinal degeneration in these conditions? I suggest that the authors discuss the implications of these findings in relation to the potential involvement of other spaceflight-related factors.
7. The authors should highlight that the use of AG represents a preventive strategy to mitigate SANS. Since the animals were placed under AG conditions after arriving at the ISS, it would be beneficial to explicitly frame the study in this context.
8. In the Materials and Methods section, I recommend that the authors describe how the animals were transported to the ISS. Additionally, it is crucial to explain how potential confounding factors, such as the stress of transportation to and from the ISS, were controlled. A discussion on how these factors were accounted for, and whether they may have influenced the transcriptomic changes observed, would help strengthen the manuscript's conclusions.
9. In the Materials and Methods section, in “mouse habitat unit-8 section”, the authors should specify why only one gender (male mice) was used for the spaceflight experiment (line 266), and whether this was consistent for the HGC mice as well (line 278).
10. In material and methods, in “mouse habitat unit-8 section”, could the authors specify when the mice were placed in the “JAXA Habitat Cages Units” and settled in continuous AG after arriving the ISS? It may help to interpretate how long time were the animals under 0G.
Minor comments
10. Reference List: Please revise the reference list, as it appears that references 3, 4, 7, and 9 have been split into two paragraphs.
11. Abbreviation Clarification: Please spell out the full name for “4-HNE” (4-hydroxynonenal) when it first appears in the text (line 188) for clarity.
12. Relevant References:
PMID: 37374046. Impact of Microgravity and Other Spaceflight Factors on the Retina of Vertebrates and Humans In Vivo and In Vitro. 2023.
PMID: 32047839. Spaceflight Associated Neuro-ocular Syndrome (SANS) and the Neuro-ophthalmologic Effects of Microgravity: A Review and an Update. 2020.
Reviewer 3 Report
Comments and Suggestions for Authors
In the article Artificial Gravity Attenuates the Transcriptomic Response to 2 Spaceflight in the Optic Nerve and Retina, Kremsky et al. presented the effects of microgravity on the retina and optic nerve, while also providing evidence of the protective role of dose-dependent artificial gravity on these structures. Their work is very interesting and the overall research could help in preventing damage of the optic nerve and retinal tissues in astronauts.
Minor comments
Line 25 in abstract: “… that mimic partial or full Earth-level gravity” could be rephrased with “that partially or fully mimic Earth-level gravity”
Line 274 – smaller font size.
In my opinion, the paper is suitable for International Journal of Molecular Sciences and is highly worth of publication. One aspect I believe could be addressed to further clarify some aspects in the discussion section is what do the authors believe could happen if the mice had been exposed for a longer amount of time in those conditions.
Round 2
Reviewer 2 Report
Comments and Suggestions for Authors
After reviewing the updated manuscript, the authors have successfully addressed most of the concerns and suggestions raised during the initial review process. They have provided the necessary clarifications for the experiments that cannot be performed and include representative images in the figure, thereby enhancing the clarity, coherence, and overall quality of the manuscript. I have no further major revisions to propose, but I do have a couple of suggestions for further improvement:
1. Although the images for 4-HNE in 0G and 0.33G show signal in the GCL. The morphology of the signals do not seem to belong to neurons in the GCL (retinal ganglion cells or displaced amacrine cells), which resembles astrocyte or most likely the end feet of the Muller cells, which used to get reactive in the pathological process (PMID: 27832276, 35804043). And although immunostaining with specific markers, such as, GFAP (astrocytes), Vimentin (Muller cells) would be recommendable, if the 4-HNE expression is in glial cells and not in neurons, could the authors discuss what the implications could be? E.g., AG may prevent the retinal gliosis associated with spaceflight.
2. Additionally, it would be recommendable including a scale bar for images in Fig 6A.
